# The mTOR Signaling Pathway in Multiple Sclerosis; from Animal Models to Human Data

**DOI:** 10.3390/ijms23158077

**Published:** 2022-07-22

**Authors:** Aigli G. Vakrakou, Anastasia Alexaki, Maria-Evgenia Brinia, Maria Anagnostouli, Leonidas Stefanis, Panos Stathopoulos

**Affiliations:** First Department of Neurology, Medical School, National and Kapodistrian University of Athens, V Sofias 72, 11528 Athens, Greece; avakrakou@gmail.com (A.G.V.); alexakh.anastasia@gmail.com (A.A.); mariaevgeniabr@gmail.com (M.-E.B.); managnost@med.uoa.gr (M.A.); lstefanis@bioacademy.gr (L.S.)

**Keywords:** mTOR, multiple sclerosis, rapamycin, metformin

## Abstract

This article recapitulates the evidence on the role of mammalian targets of rapamycin (mTOR) complex pathways in multiple sclerosis (MS). Key biological processes that intersect with mTOR signaling cascades include autophagy, inflammasome activation, innate (e.g., microglial) and adaptive (B and T cell) immune responses, and axonal and neuronal toxicity/degeneration. There is robust evidence that mTOR inhibitors, such as rapamycin, ameliorate the clinical course of the animal model of MS, experimental autoimmune encephalomyelitis (EAE). New, evolving data unravel mechanisms underlying the therapeutic effect on EAE, which include balance among T-effector and T-regulatory cells, and mTOR effects on myeloid cell function, polarization, and antigen presentation, with relevance to MS pathogenesis. Radiologic and preliminary clinical data from a phase 2 randomized, controlled trial of temsirolimus (a rapamycin analogue) in MS show moderate efficacy, with significant adverse effects. Large clinical trials of indirect mTOR inhibitors (metformin) in MS are lacking; however, a smaller prospective, non-randomized study shows some potentially promising radiological results in combination with ex vivo beneficial effects on immune cells that might warrant further investigation. Importantly, the study of mTOR pathway contributions to autoimmune inflammatory demyelination and multiple sclerosis illustrates the difficulties in the clinical application of animal model results. Nevertheless, it is not inconceivable that targeting metabolism in the future with cell-selective mTOR inhibitors (compared to the broad inhibitors tried to date) could be developed to improve efficacy and reduce side effects.

## 1. Introduction

### 1.1. Multiple Sclerosis—Basic Aspects of Pathophysiology

MS is an autoimmune, inflammatory, and neurodegenerative demyelinating disorder of the central nervous system (CNS), and the leading cause of non-traumatic disability in young adults [1]. Pathologic studies reveal (1) demyelination and varying degrees of remyelination [2,3,4], (2) acute inflammation characterized by myelin-laden macrophages and lymphocytes [5], (3) chronic inflammation at the rim of demyelinating lesions with glial activation and at the meninges with lymphocytic infiltrates, sometimes in the form of lymphoid follicles [6,7], and (4) axonal and neuronal damage [8,9,10]. The pathology involves the white as well as the gray matter [11]. Although the pathogenesis is not fully understood, a multifactorial etiology hypothesis was proposed including associated genetic risk loci [12,13], and multiple environmental variants such as low vitamin D [14], EBV infection, obesity, and smoking [15]. The clinical course of the disease is highly variable. In most patients (>80%), MS initiates as a relapsing/remitting disease (RRMS), with partial or complete recovery in between relapses. The frequency of relapses gradually decreases, and most patients enter a phase of gradual progression of neurologic disability (secondary progressive MS, SPMS). In 10–15% of cases, the relapsing phase is missing, and the disease is progressive from the onset (primary progressive MS, PPMS) [16].

The process leading to CNS lesions involves myelin-reactive CD4 T lymphocytes, especially the T helper 1 (Th1) and Th17 subsets [17], which can cross the blood–brain barrier (ΒΒΒ), infiltrate the CNS, and promote the inflammatory response. The recruitment of other peripheral immune cells (including myeloid, CD8 T cells, B cells, and natural killer cells), and parenchymal glial cells such as microglia, further sustains this response [18,19,20]. Strong evidence supports the involvement of B cells in MS pathogenesis with various mechanisms, such as cytokine (IL-10, IL-6, lymphotoxin α, GM-CSF, TNF-a) production (‘bystander activation or suppression’), antigen processing for presentation to T cells, and possibly antibody production [21,22]. Importantly, B cells were found to infiltrate the affected brain tissue of MS patients, and a number of B cell subtypes, especially memory B cells and plasmablasts, were found to be present in the cerebrospinal fluid (CSF) of MS patients [23]. Immune effector mechanisms more associated with progressive forms of MS include ectopic formation of B cell follicle-like structures, and microglial and astrocytic activation [24,25,26]. Innate immune cells of the CNS, such as microglial cells and peripheral blood-derived monocytes (known as monocyte-derived macrophages), play a key role in MS pathogenesis, and can directly affect demyelination (macrophages) and promote the accumulation of axonal damage [27,28]. Early acute MS lesions are mainly composed of dense activated microglia and macrophages throughout the entire lesion [29]. In addition, macrophages/activated microglia at the rim of slowly expanding, chronic active MS lesions are characterized by a loss of the homeostatic microglial signature, and predominant expression of molecules involved in phagocytosis, oxidative injury, antigen presentation, and T cell co-stimulation [30,31].

The treatment modalities currently used for MS are immunomodulatory and immunosuppressive; these drugs are more effective in the early phases of the disease (RRMS), where acute inflammation is more prominent, but are probably unable to target chronic inflammation and axonal loss, which present, to a greater extent, during the progressive phase. Furthermore, at this stage, inflammation is mostly compartmentalized within the CNS behind an intact or almost intact BBB, which is not sufficiently permeable to most existing drugs (e.g., monoclonal antibodies) [32]. Moreover, existing therapies cannot stimulate remyelination [33]. It is, therefore, imperative to discover drugs that can cross the BBB, target both innate and adaptive immunity, stimulate remyelination, stop axonal loss, and reduce disability progression. In this context, signaling pathways regulated by mammalian target of rapamycin (mTOR) are described as playing an important role in the regulation of both innate and adaptive immune responses. In the present article, we review all currently available evidence on the role of the mTOR pathway in MS, and speculate on its possible therapeutic use.

### 1.2. mTOR as a Master Regulator of Growth, Metabolism, and Survival

mTOR is a conserved serine/threonine-protein kinase that belongs to the phosphatidylinositol 3-kinase-related kinase family. mTOR plays an important role in the signaling network that regulates growth and metabolism in response to environmental cues such as glucose, amino acids, growth factors, and WNT (Wingless)/β-Catenin signaling (Figure 1), as well as in a wide array of biological processes including cell cycle control [34], protein synthesis, energy balance, proliferation, survival [35], and cellular senescence [36]. In mammals, mTOR constitutes the catalytic subunit of two distinct complexes known as mTOR complex 1 (mTORC1) and mTOR complex 2 (mTORC2). These complexes are distinguished by their accessory proteins and their differential sensitivity to rapamycin, as well as somewhat different upstream signals, substrates, and cellular functions, but share promoting anabolic and inhibiting catabolic processes [37,38].

mTORC1 exists as a multiprotein complex containing mTOR, regulatory-associated protein of mTOR (Raptor), inhibitory protein Deptor, inhibitory protein PRAS40, and adaptor protein mLST8. Raptor positively regulates mTOR activity, and acts as a scaffold for the recruitment of mTORC1 substrates. mTORC1 is activated in response to different intracellular and extracellular clues, while including growth factors, cytokines, energy status, oxygen, and amino acids. mTORC1 is involved in a variety of biological processes, including autophagy, lipid synthesis, and mitochondrial metabolism. PI3K–AKT is the main upstream signaling pathway, and its activation converges on the tuberous sclerosis complex (TSC), consisting of TSC1 and TSC2 proteins that inhibit mTORC1 activation (Figure 1). Extracellular signals serve as the regulation of multiple functions related to cell-cycle progression and cellular growth, and include the activation of different receptors by hormones, cytokines, and growth factors that enhance mTORC1 activity. In addition to extracellular stimuli, mTORC1 is responsive to internal signals (energy status, oxygen, and amino acids). For example, when cellular energy is low, the AMP kinase (AMPK) blocks mTORC1 activity, via the activation of the inhibitory protein TSC2. The activity of mTORC1 is also sensitive to nutrient availability. Increasing the cellular concentration of amino acids, such as leucine, activates mTORC1. The activation of mTORC1 generally increases the cellular capacity of protein generation. The two main downstream targets of mTORC1, the eukaryotic initiation factor 4E-binding protein 1 (4E-BP1) and the ribosomal S6 protein kinase 1 (p70S6K), are key components of the protein translation machinery [35].

mTORC2 comprises mTOR, adaptor protein Rictor, mSIN1, Deptor, mLST8, and Protor. In contrast to mTORC1, for which upstream signal and cellular functions were identified, relatively little is known about mTORC2, due to the absence of effective mTORC2 inhibitors. Although the signaling pathways that lead to mTORC2 activation are not well characterized, growth factors and PI3K are also considered signals for regulating this pathway [37]. mTORC2 phosphorylates and activates AKT kinase to further promote nutrient uptake and cell survival. mTORC2 is mainly involved in the regulation of phosphorylation, and the activation of AKT/PKB, protein kinase C, and serum- and glucocorticoid-induced protein kinase 1 (SGK1). There is complicated cross-talk between mTORC1 and mTORC2. AKT is an upstream activator of mTORC1 via the phosphorylation and inhibition of TSC1/2, while TSC1/2 positively regulates mTORC2 to indirectly regulate AKT [39]. Furthermore, the S6K (a downstream target of mTORC1) negatively modulates mTORC2 activity [40]. Therefore, mTORC1 and mTORC2 signaling are finely tuned to respond to dynamic changes in metabolism and environmental signals.

### 1.3. mTOR Inhibitors

Drugs targeting the PI3K/AKT/mTOR pathway are under investigation for their potential use in autoimmunity as immunosuppressant agents (Table 1). The prototypic mTORC1 inhibitor rapamycin is a macrolide immunosuppressant isolated from Streptomyces hygroscopicus, was successfully used to prevent allograft rejection in transplantations [41]. Rapamycin is considered a classical autophagy inducer and a specific inhibitor of mTOR by regulating autophagy-related proteins and lysosome biosynthesis. The drug mainly inhibits mTORC1 activity by disrupting the association with Raptor. Conversely, mTORC2 activity is resistant to short-term treatments with rapamycin. Due to its lipophilic nature, rapamycin easily crosses the BBB, allowing it to exert direct effects within the CNS [42].

A second way of achieving mTORC1 and mTORC2 inhibition is by AMP-activated protein kinase (AMPK) activation. When cellular energy is low, the AMPK blocks mTORC1 activity by increasing the inhibitory activity of TSC2. Moreover, AMPK can directly modulate the activity of mTORC2 by the phosphorylation of this complex, thus, blocking its nuclear translocation and regulating gene transcription. Of medical relevance, the antidiabetic drug metformin was found to enhance AMPK activation both in vitro and in vivo.

mTOR inhibitors were considered as possible candidates for the treatment MS, and were tested in EAE animal models of MS. In EAE, inflammatory demyelination can be induced by the immunization of several animal species (mice, rats, marmosets) with myelin antigens (or CNS tissue), or by adoptive–transfer of pathogenic, myelin-specific CD4 T cells generated in donor animals by active immunization. EAE better simulates “efferent” pathways of MS, and is widely used to study the immunopathologic mechanism of MS, as well as possible therapeutic interventions; however, it does not model “afferent” pathways well (induction of disease).

Rapamycin administration is repeatedly shown (Table 2) to inhibit the induction of EAE and the progression of established EAE [43,44,45,46,47,48,49,50,51]. Moreover, T-cell-targeted knockout of insulin-like growth factor 1 receptor (IGFR1) (an mTOR activator) ameliorates EAE in mice [46].

The cuprizone model, a model entirely different from EAE, involves feeding animals a diet containing cuprizone. Cuprizone causes consistent demyelination in both the white and gray matter within the CNS, and is toxic to mature oligodendrocytes without causing direct neuronal injury [52]. In this model, rapamycin administration leads to conflicting results (reported both as beneficial and detrimental), while metformin administration and oligodendrocyte precursor cell (OPC)-specific mTOR knockout is shown to have a beneficial effect [53,54,55,56].

**Table 2 ijms-23-08077-t002:** mTOR inhibitors or knockout in animal of multiple sclerosis.

Animal Model	Intervention	Emerging Role of mTOR Pathway Manipulations	Change in Disease Status	Ref.
RR-EAE (SJL/j mice, PLP_139–151_ peptide)	Rapamycin	↓ Teff, ↑ Tregs	Improvement	[43]
RR-EAE (SJL/j mice, PLP_139–151_ peptide) and adoptive transfer of encephalitogenic T cells (donor cells from SJL/j mice immunized with PLP_139–151_)	tNPs; polymers that encapsulate peptide Ag and rapamycin	↑ Ag-specific Tregs and ↓ T cell-mediated autoimmunity	Improvement	[57]
PR-EAE (DA rats, whole DA rat spinal cord)	Rapamycin	↓ splenic CD8 T cells, ↑ splenic Tregs	Improvement	[49]
M-EAE (C57BL/6 mice, MOG_35–55_)	Rapamycin	↓ Th1, ↓ Th17 cells	Improvement	[45]
M-EAE (C57BL/6 mice, MOG_35–55_)	Rapamycin	↓ Th17 cells, ↑ Tregs	Improvement	[48]
M-EAE (C57BL/6 mice, MOG_35–55_)	Rapamycin	↓ neuronal cell death, ↑ autophagy	Improvement	[44]
M-EAE (C57BL/6 mice, MOG_35–55_)	Rapamycin ex vivo post stimulation with MOG/LPS	↓ IL-17, TBX21, RORc, IFN-γ, TNF-α	Improvement	[58]
M-EAE (C57BL/6 mice, MOG_35–55_)	Rapamycin/MCC950	↓ neuronal cell death, ↑ autophagy	Improvement	[50]
M-EAE (C57BL/6 mice, MOG_35–55_)	Rapamycin/MCC950	Restoration of gut microbiota, ↑ autophagy	Improvement	[51]
M-EAE (C57BL/6 mice, MOG_35–55_)	AZD8055	↓ inflammation, ↑ autophagy	Improvement	[59]
M-EAE (C57BL/6 mice, MOG_35–55_)	Rapamycin	↑ TAM receptors, ↑ anti-inflammatory factors and ↓ TLR-3, TLR-4	Improvement	[60]
M-EAE (C57BL/6 mice, MOG_35–55_)	Rapamycin and fingolimod	↓ Th17 and ↑ Treg cells in the spleen and CNS	Improvement	[45]
C-EAE (C57BL/6 mice, MOG_35–55_)	HSO/EPO and/or rapamycin	↑ regeneration of the myelin sheath	Improvement	[61]
C-EAE (C57BL/6 mice, MOG_35–55_)	HSO/EPO	↓ STAT3 and IL-17 genes and ↑ FOXP3+ gene in spinal cord	Improvement	[62]
C-EAE (C57BL/6 mice, MOG_35–55_)	HSO/EPO and/or rapamycin	↓ RAPTOR, IFNγ, IL-17, STAT3 and ↑ RICTOR, IL-10, FOXP3 genes	Improvement	[63]
Cuprizone (C56BL/6 J mice, 0.3% CPZ diet)	Rapamycin	↓ spontaneous remyelination	Worsening	[54]
Cuprizone (C57BL/6 J mice 0.2% CPZ diet)	Metformin	↑ myelin recovery in oligodendrocytes	Improvement	[53]
Cuprizone (Ng2-Mtor cKO of C57Bl/6 background, 0.2% CPZ diet)	OPC-specific mTOR KO	Delay in myelin production	Delayed improvement	[55]
Cuprizone (C57BL/6 J mice 0.2% CPZ diet)	Rapamycin	↑ inflammation and callosal axonal damage	Worsening	[56]

Abbreviations: ↓/↑: decrease/increase level; EAE: experimental autoimmune encephalomyelitis; RR-EAE: relapsing–remitting EAE; PR-EAE: protracted relapsing EAE; C-EAE: chronic EAE; DA: Dark Agouti; M-EAE: monophasic EAE; KO: knockout; tNPs: tolerogenic nanoparticles; Ag: antigen; MCC950: an inflammasome inhibitor; mTOR: mammalian target of rapamycin; OPC: oligodendrocytes precursor cell; Ng2: neural/glial antigen 2; Teff: effector T; Tregs: T-regulatory cells; Th: T-helper cell; MOG_35–55_: myelin oligodendrocyte glycoprotein p_35–55_; CPZ: cuprizone; TAM: Tyro3, Axl, and Mer protein tyrosine kinase receptors; TLR: toll-like receptors; CNS: central nervous system; HSO/EPO: hemp seed/evening primrose oils; IL-10: interleukin-10; IL-17: interleukin-17; FOXP3: forkhead box P3; RAPTOR: regulatory-associated protein of mTOR; RICTOR: regulatory-associated companion of mTOR; IFNγ: interferon gamma; STAT3: signal transducer and activator of transcription factors.

## 2. Specific Role of mTOR on the Biological Process of Autophagy and Its Relevance to CNS Demyelination and MS

Autophagy is an intracellular catabolism pathway system responsible for the degradation of proteins, damaged organelles, pathogens, and, thus, maintains cell homeostasis [64]. Various intracellular components are sequestered within double-membrane vesicles, also called autophagosomes, which subsequently fuse with lysosomes (autophagolysosomes) [65]. Many conditions inducing autophagy, such as nutrient, cellular energy, or growth factor deprivation, are shown to inhibit mTORC1 activity, suggesting an inverse relationship between autophagy induction and mTORC1 activation. Many critical steps of autophagic pathways are regulated by mTOR, such as the activation of the beclin-1 complex and LC3-II [66].

A dual role of autophagy has been described in MS and EAE pathogenesis, among cell subsets. On one hand, in T cells, enhanced levels of autophagy are associated with increased antigen presentation of myelin to CD4 autoreactive T cells [67] and elevated expression of autophagy-related 5 protein (Atg5), with deterioration of MS [64]. On the other hand, defective autophagy in neurons leads to elevated reactive oxygen species (ROS) levels and impaired clearance of damaged mitochondria and myelin debris by microglia, promoting a detrimental pro-inflammatory phenotype involving inflammasome activation [67]. Likewise in glial cells, astroglial autophagy inhibition with small interfering RNA against autophagy-related 5 protein (Atg5) induces neuronal death [67]. In a similar context, autophagy-deficient oligodendrocytes lead to a reduction in the number of myelinated axons, and a decreased thickness of the myelin sheath [68,69,70] (Figure 2). Thus, the effects of autophagy are cell type-specific, and dependent on whether the autophagic pathway is turned on or off, with increased autophagy being detrimental in immune cells promoting autoimmunity, but beneficial for neurons, astrocytes, and oligodendrocytes.

Several settings in in vivo studies indicate that the protective effects of autophagy in neurons, astrocytes, and oligodendrocytes may be more significant than the effect on antigen presentation. In addition to all EAE experiments with autophagy inducers/mTOR inhibitors such as rapamycin (Table 2), caloric restriction is shown to increase autophagy and, at the same time, improve disease severity by stimulating remyelination in EAE and cuprizone-fed mice [71]. Not surprisingly, selective inhibition of autophagy in antigen-presenting cells (while sparing autophagy in neurons, astrocytes, and oligodendrocytes) also has a beneficial effect in EAE. Dendritic cell (DC)-specific deletion of ATG-7 results in defective autophagy and defective antigen presentation [72]. Further, myeloid cell-specific Pik3c3 (subunit of the class III phosphatidylinositol 3-kinase complex, regulator of autophagy) knockout leads to decreased myeloid cell-derived IL-1β production and myelin-specific CD4 T cells levels in the brain [73].

In humans, immunopathological examinations show the presence of elevated autophagy markers in various compartments of RRMS and SPMS patients, likely reflecting increased autophagy of autoreactive immune cells. Specifically, studies in MS patients show elevated levels of synaptic vesicle-containing autophagosomes in the dentate nucleus [74], Atg5 expression in encephalitogenic T cells in biopsies [64], and both ATG5 and parkin (mitophagy marker) in the cerebrospinal fluid (CSF) and serum [75], compared to non-disease controls. Moreover, in 95 patients with MS, gene expression levels from blood autophagy-related genes (ATG16L2, ATG9A, BCL2, FAS, GAA, HGS, PIK3R1, RAB24, RGS19, ULK1, FOXO1, HTT) were significantly altered (false discovery rate < 0.05, either up or downregulated) compared to healthy individuals [76]. It should be noted, however, that, apart from autophagy, increased mTOR activation (e.g., as shown by increased ribosomal protein S6 expression) is also shown in MS patients [77].

### 2.1. Role of mTOR in Innate Immune Responses in Myeloid, Microglial Cells and Inflammasome

The effects of mTOR on the immune system can be divided into effects on innate immunity and effects on the adaptive immune response. In addition to autophagy effects on antigen presentation, the mTOR pathway can integrate signals to orchestrate lineage differentiation of myeloid cells (Figure 2). Specifically, mTOR signaling has been linked to macrophage polarization [78]. M2-type polarization (considered immunoregulatory) involves AMPK/mTORC2 signaling upregulation that leads to fatty acid oxidation (FAO) through peroxisome proliferator-activated receptor beta coactivator 1 (PGC-1β) [79,80], and also requires both mTORC1 and late endosomal/lysosomal adaptor and MAPK and mTOR activator (LAMTOR or ragulator) [81]. In accordance, it is shown that the PI3K–mTOR network promotes M2 macrophage polarization and, thus, rapamycin-treated human macrophages show a proinflammatory M1-polarized phenotype [82]. In microglial cultures, mTOR inhibition affects microglial polarization by inducing M1-polarization, promoting NOS2 and cyclooxygenase pathways [83], while rapamycin-induced autophagy suppresses the expression of iNOS, IL6, and cell death of LPS-stimulated microglia.

In some contrast, mTOR inhibition is also shown to decrease the inflammatory phenotype of DCs, which are of myeloid lineage, and promotes tolerogenic effects [84]. Moreover, inhibition of mTOR with rapamycin is associated with the inhibition of inflammasome activation and improved clinical presentation of EAE; these effects are mainly mediated by the inhibition of microglial activation [50]. In further support, AZD8055, an mTOR suppressor and autophagy activator, leads to EAE amelioration through the suppression of NLRP3- and inflammasome-mediated pyroptosis [59]. It is likely that the mTOR inhibition effects on the myeloid cell differentiation program are highly dependent on the specific type of cells, their activation status, and the inflammatory milieu. In support of that notion, the activation of AKT1 and mTOR pathways (and not the inhibition with induction of autophagy) protects microglial cells from apoptotic cell death during oxidative stress through the inhibitory phosphorylation of GSK-3β that subsequently blocks genomic DNA degradation and membrane phosphatidylserine (PS) exposure [85].

### 2.2. Role of mTOR in Adaptive Immune Responses of T and B Cell Development and Differentiation

In addition to its action on the cell cycle, which enables the rapid division of cells including lymphocytes responding to an antigen, the mTOR pathway can play a critical role in shaping T cell responses by influencing T-effector cell development [86] (Figure 2). mTOR modulates T cell lineage differentiation through integrating signals derived from the T cell receptor (TCR), co-stimulatory molecules, and cytokines. Importantly, T cells devoid of mTOR are not capable of differentiation in either Th1, Th2, or Th17 cells [87,88,89]. Specific proteins of the mTOR complex display distinct functions in T cell differentiation programs. mTORC1 is implicated in Th1 cell differentiation, and mTORC2 is required for Th2 differentiation. The loss of both mTORC1 and mTORC2 activities enhances the differentiation of regulatory T cells (Tregs) [86]. Further, loss of LAMTOR (the mTOR activator) leads to impaired Th17 polarization, and T cell-specific LAMTOR knockout mice are protected against EAE [90]. Not surprisingly, mTOR inhibition through the action of metformin restricts T-effector cells and promotes T-regulatory cells [89]. Moreover, mTOR inhibitors such as rapamycin promote T cell anergy, while mTOR activation leads to the opposite direction by generating CD8 memory T cells [91,92]. Collectively, mTOR inhibition blocks T-effector development and function, and can limit related autoimmunity. With that in mind, the protective effect of rapamycin in EAE can be explained both with the effect on T cells, in addition to the aforementioned possible protective effect on neurons, astrocytes, and oligodendrocytes connected to autophagy induction.

mTORC signaling is also present in B cells, and plays a crucial role in their development and function [93] (Figure 2). Basic metabolic needs of naive B cells are supplied by OXPHOS and glycogen synthesis, whereas mTORC1-dependent glutaminolysis supports the proliferation/clonal expansion of memory B cells and plasma cells (as well as of many differentiated T cell subtypes) [94,95]. B cell receptor (BCR) signaling can activate PI3K and mTOR and, in high PI3K/AKT conditions, direct B cells towards an unswitched plasma cell fate [95,96]. In addition, mTORC1 signaling is required for germinal center function; specifically the pathway is triggered by T cell help, and promotes the anabolic process of dark zone centroblast proliferation [97,98]. mTOR inhibition with rapamycin inhibits B cell-, BCR- and toll-like receptor-mediated mTOR signaling in B cells, profoundly limiting their proliferation and survival [99,100]. Interestingly, hyperactive B cell mTORC1 signaling is seen in models of autoimmunity (in particular lupus) [101]. Overall, and similarly to T cells, mTOR activation is associated with B cell differentiation and function (including autoimmune settings), while its blockade restricts it.

However, there is another side to mTOR, autophagy, B cells, and autoimmunity. As B cells can present antigen and autophagy, linked to antigen presentation [102], autophagy inducers could also promote autoimmunity, as in the case of citrullinated peptides, which are relevant in rheumatoid arthritis [103]. Further, autophagy of B cells is found to increase in murine models of lupus even prior to disease onset [104], and this increase is greater in immature B cells; autophagy of B cells is also increased in patients with systemic lupus erythematosus. Based on these data, induction of autophagy with an mTOR inhibitor could be detrimental in autoimmune settings [102,103,104].

## 3. Human Data Regarding mTOR Inhibition

In the last decades, a big effort was made to investigate the pathogenesis of MS, in order to study and develop new treatment approaches that maximize efficacy and minimize adverse reactions. Rapamycin (also termed sirolimus) can be considered one such candidate, due to its efficacy in EAE. In addition, the group of mTOR inhibitors similar to rapamycin is growing, and includes the so-called rapalogs such as CCI-779 (temsirolimus), RAD-001 (everolimus), and AP23573 (ridaforolimus or deforolimus). Despite the extensive EAE literature, reports of mTOR inhibitor effects in MS patients are limited, and include small clinical trials and cohort studies, and a phase 2 study from 2005, which was only published as an abstract of a European neurological society meeting (Table 3).

In a small trial of six patients with RRMS that received 2 mg of daily rapamycin for 6 months and six healthy controls, a reduction in the pro-inflammatory IFN-γ and an increase in the regulatory TGF-β post rapamycin is noted in the serum of patients compared to controls [105]. In further investigations of eight patients with RRMS treated with 2 mg of daily rapamycin for 6 months without a control group, a significant decrease in mean and maximum MRI lesional area is noted, along with an increase in Foxp3 Tregs at the end of the 6 month period compared to before treatment (no comparisons to the control group are reported). Clinically, four out of eight patients show a reduction in neurological disability; however, this reduction is non-significant [106]. Overall, the low number of patients, the largely uncontrolled nature of the study, the low dosage of the drug, and the short follow-up preclude meaningful conclusions from these investigations. Unfortunately, a phase I/II open-label pilot trial that evaluated the safety of sirolimus in 14 patients with RRMS was terminated in 2016, and no results were reported (NCT00095329).

The best data on the application of rapamycin/rapalogs in MS include radiologic and preliminary clinical results from a 2005 phase-II clinical trial. The trial studied the effects of three different doses of temsirolimus (2, 4, and 8 mg) vs. placebo in 296 patients with active, relapsing MS for a period of 9 months, with monthly MRI scans. The 8 mg dose is found to significantly reduce active lesions and relapses by approximately 50% compared to placebo [107,108,109]. Side effects are not negligible, and include mouth ulcerations and aphthous stomatitis, menstrual dysfunction, hyperlipidemia, and rashes (increased vs. placebo). The overall safety and risk/benefit ratio prevented advancement to phase 3 trials.

In regard to metformin, a prospective study that included 20 obese RRMS patients with metabolic syndrome treated with metformin (850–1500 mg/day) for 6 months and followed for a mean of 26.7 months, as well as 10 patients sharing the same characteristics treated with pioglitazone, and 20 untreated control patients, reveals a significant reduction in the number of new or expanding T2 lesions and gadolinium-enhancing lesions in the metformin-treated group compared to the control group. A similar reduction is noted in the pioglitazone group. In addition, a significant reduction in myelin-specific T cells and an increase in Tregs is noted in both treatment arms; however, there is no change in disability or relapse rate. These interesting results could be confounded by the metabolic syndrome, but warrant further investigation [110].

**Table 3 ijms-23-08077-t003:** Studies of mTOR inhibitors in multiple sclerosis patients.

Type of Study	Patients Included	Main Results	Ref.
Double-blind, placebo-controlled phase II trial	N = 296 patients with active, relapsing MS relapses; 2, 4, 8 mg temsirolimus/d or placebo for 9 months	8 mg group: ~50% reduction in active lesions and relapses, mouth ulcers, menstrual abnormalities	[107,108]
Non-randomized, prospective, controlled	N = 50 RRMS with metabolic syndrome; N = 20 metformin, N = 20 untreated	Metformin group:↓ new/expanding T2 lesions, ↓ Gd+ lesions, ↓ myelin-specific T cells, ↑ Tregs	[110]
Non-randomized, prospective, controlled	N = six RRMS patients on 2 mg rapamycin/d for 6 months N = six healthy controls	↓ IFNγ, ↑ TGFβ in serum of patients after 6 months	[105]
Non-randomized, prospective, uncontrolled	N = eight RRMS patients on 2 mg rapamycin/d for 6 months	↑ Tregs, ↓ in mean lesional area size after treatment	[106]

Abbreviations: ↓/↑: decrease/increase level; Gd+: gadolinium-enhancing lesions; d: day; IFNγ: interferon gamma; MS: multiple sclerosis; mTOR: mammalian target of rapamycin; RRMS: relapsing–remitting MS; TGFβ: transforming growth factor beta; Tregs: T regulatory cells.

## 4. Conclusions

As examined throughout this review, evidence from animal models and humans supports the notion that the mTOR network may have significant involvement in the pathogenesis of MS. mTOR signaling pathways influence the cell cycle and cell division, but are also closely related to autophagy mechanisms in various cell types, including immune cells of innate and adaptive immunity (well-known to participate in MS disease mechanisms), as well as neurons and glial cells. mTOR cell-type specific functions under pathological or physiological conditions are the subject of research and some debate. Rapamycin- or metformin-induced mTOR inhibition, and the consequent autophagy induction, can function homeostatically, and protect neurons and glial cells from ROS and damage leading to cell death. Although autophagy is largely involved in the resolution of inflammation, and resetting it to homeostasis [111], increased myelin antigen presentation to CD4 T cells due to elevated autophagic function may produce a dysregulated pathological immune response. Moreover, mTOR inhibition promotes pro-inflammatory M1 macrophage polarization. Interestingly, the opposite effects of mTOR inhibition is also observed in dendritic and microglial cells. In these experiments, mTOR inhibition suppresses the inflammasome and reduces inflammatory activity. At the same time, inhibition of mTOR by rapamycin or metformin in T cells leads to an increase in Tregs, T cell anergy, and T-effector decline in in vivo studies. Notably, the development of B cells is also an mTOR-dependent process, including their survival and differentiation into memory B cells and plasma cells, which is decreased in the presence of rapamycin. However, excessive B cell mTORC1 signaling is detected as a prevalent factor in mice models of autoimmunity, and elevated autophagy due to mTOR inhibition, along with antigen presentation by B cells, appears to amplify autoimmunity in mice models.

In vivo models of mTOR inhibition, where these opposing actions take effect simultaneously, demonstrate that mTOR inhibitors (rapamycin, metformin) ameliorate EAE. In humans, clinical trials with mTOR inhibitors (rapamycin/sirolimus, temsirolimus, metformin) aimed to assess potential beneficial effects for patients with MS, both clinical and mechanistic, the latter by examining different cell types and cytokines implicated in MS pathophysiology. The most remarkable preliminary (phase 2) clinical data to date show that temsirolimus reduces both active lesions and relapses in relapse MS patients by half compared to placebo; however, the side effects are not negligible, and the risk/benefit ratio prevented the pursuit of phase 3 trials. In this context, the field of mTOR inhibitor therapy aims both at drugs that can more effectively eliminate pathogenic mTOR pathways of MS, but also display reduced side effects.

Collectively, the genetic, in vitro, and in vivo data presented in this review indicate a role for the deregulated mTOR pathway in the pathogenesis of MS. Current results from clinical trials of rapamycin and its analogues in MS have not been encouraging in terms of the risk/benefit ratio, whereas data for metformin are limited. Further studies in larger populations and randomized trials could provide a clearer picture. Moreover, it is not inconceivable that targeting metabolism in the future with cell-selective mTOR inhibitors (compared to the broad inhibitors tried to date) could be developed to improve efficacy and reduce side effects. Overall, we can appreciate that a deeper understanding of how mTOR activity contributes to disease pathogenesis is needed in order to harness mTOR therapeutic potential.

## Figures and Tables

**Figure 1 ijms-23-08077-f001:**
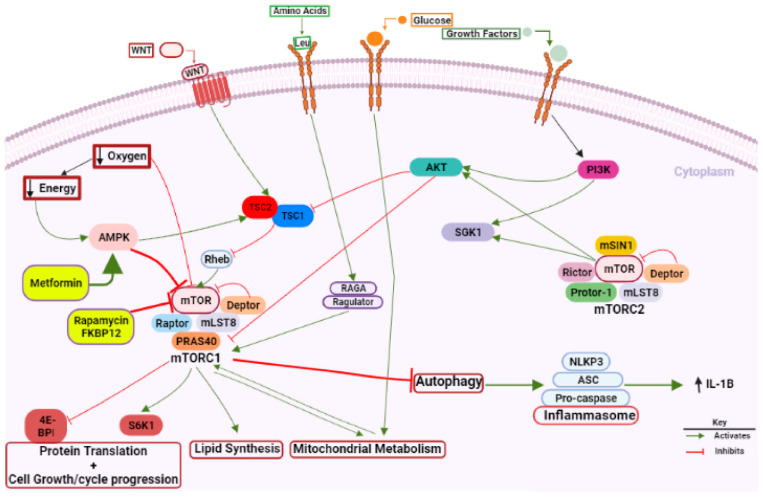
mTOR signaling pathway. mTOR is activated by the presence of growth factors, glucose, amino acids, WNT at the cell-surface, and by several intracellular clues. Signaling to mTORC1 is stimulated by growth factors, WNT via PI3K/AKT signaling. In an environment with sufficient nutrients, the autophagy process is negatively regulated by mTOR. AMPK activation, starvation, or hypoxia inhibit mTOR. Rapamycin, the inhibitor of mTOR, represses autophagy, and regulates inflammation. There is cross-talk between inflammasomes and autophagy to maintain homeostasis among the inflammatory response against pathogens and inhibition of disproportionate inflammation. During mTOR inhibition of autophagy, NLRP3 inflammasome activation is enhanced, and this effect is reversed after rapamycin treatment. Metformin enhances autophagy and inhibits NLRP3 inflammasome via AMPK/mTOR signaling inhibition. mTOR is implicated in several biological processes: protein translation, cell growth/cycle progression, lipid synthesis, and mitochondrial metabolism. Created with BioRender.com. 4EBP1: eukaryotic translation initiation factor 4E-binding protein 1; AMPK: AMP-activated protein kinase; ASC: C-terminal caspase recruitment domain; mTOR: mammalian target of rapamycin; NLRP3: nucleotide-binding oligomerization domain (NOD)-like receptor protein 3; RAG: regulator; RHEB: ras homolog enriched in brain; S6 Kinase 1: ribosomal protein S6 kinase beta-1; TSC: tuberous sclerosis complex. Figure created with BioRender.com (last accessed on 16 July 2022).

**Figure 2 ijms-23-08077-f002:**
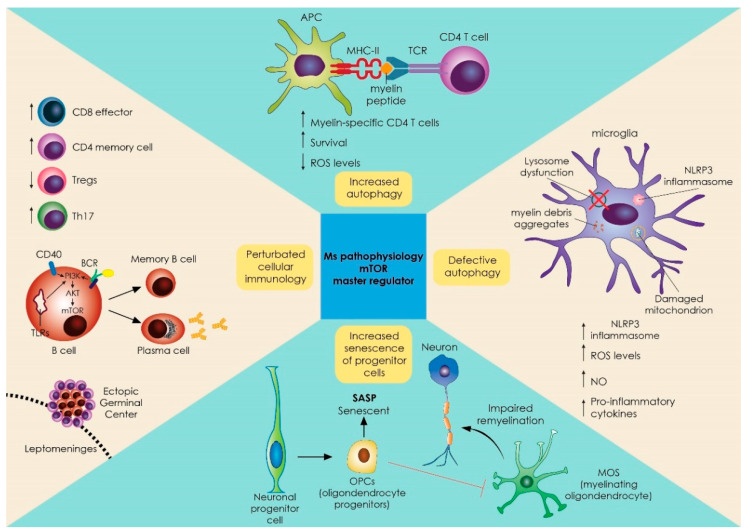
mTOR signaling pathway as a master regulator of many aberrant immune responses and biological processes implicated in MS pathogenesis. mTOR is critical in the development and function of various immune cells that play an important role in MS pathology. The picture shows the effects of mTOR inhibition in T cells, B cells, oligodendrocyte progenitor cells, macrophages, and microglial cells, as well as the effects of mTOR on the autophagy process that regulates the inflammatory phenotype of various cells such as neurons, microglia, and other glial cells. In the top part, increased levels of autophagy could lead to enhanced myelin processing, and antigen presentation to CD4 autoreactive T-cells in antigen presenting cells. Increased autophagy also prolongs survival of activated CD4 and CD8 T-cells. On the right, defective autophagy leads to aberrant clearance of damaged mitochondria, inflammasomes, and myelin debris in microglia, and fuels a pro-inflammatory phenotype. At the bottom of the figure, neural progenitors are affected by the presence of SASP, affecting the differentiation of OPCs and reducing remyelination of neurons. On the left, upstream regulation of BCR, CD40, and TLR signaling leads to a downstream PI3K/AKT/mTOR pathway and differentiation of B cells by T cell help, promoting the anabolic process of dark zone centroblast proliferation. AKT: protein kinase B; APC: antigen-presenting cell; BCR: B cell receptor; Th: T helper cell; PI3K: phosphoinositide 3-kinase; MHC: major histocompatibility complex; mTOR: mammalian target of rapamycin; NLRP3: nucleotide-binding oligomerization domain (NOD)-like receptor protein 3; NO: nitric oxide; ROS: reactive oxygen species; SASP: senescence-associated secretory phenotype; T reg: T-regulatory cells.

**Table 1 ijms-23-08077-t001:** A representative list of mTOR inhibitors.

Inhibitor	Category of mTOR Inhibitor	Inhibition Effect
Rapamycin (AY-22989)	Rapamycin	Direct mTOR inhibition
Temsirolimus (CCI-779)	Rapalogs	
Everolimus (RAD-001)		
Ridaforolimus (AP23573)		
OSI-027	TORC1/TORC2 inhibitors	
Vistusertib (AZD2014)		
AZD8055		
Dactolisib (BEZ235)	PI3K/mTOR inhibitors	Indirect mTOR inhibition
Apitolisib (GDC-0980)		
Gedatolisib (PF05212384)		
Omipalisib (GSK2126458)		
Metformin (A10BA02)	AMPK activation	

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
