# Peer review of "The mTOR Signaling Pathway in Multiple Sclerosis; from Animal Models to Human Data"

_ijms, 2022, doi:10.3390/ijms23158077_

Round 1

Reviewer 1 Report

The review is well written, succinct and brings important information in the context of mTOR in MS. Here are some small notes:

Abstract: 

- The abstract could be improved. I see no need to expose adverse events ("such as mouth ulceration and menstrual abnormalities") and talk more about the possible mechanism of action in the disease . 

1. Introduction

1.1. Multiple sclerosis – basic aspects of pathophysiology

- The authors focus only T cells, but in MS there is also the action of B cells and resident innate immune cells (microglia). The other cells (B and innate cells) are important in MS and are also affected by mTOR. Considering that the authors included these cells in the figure 2, in important to mentioned the function of these cells in MS pathogenesis. 

- Furthermore, the authors bring data from the experimental model (EAE), but do not include a description of models in the text. This is important to the reader, as there are different models (rat, C57BL/6, SJL, passive, active), with specific characteristics that mimic or not the conditions observed in patients.

1.3. mTOR inhibitors

- Please, the name of any species must be written in italics. See line 133 for Streptomyces hygroscopicus, that is a species of bacteria.

2. Specific role of mTOR on the biological process of autophagy and its 162 relevance to CNS demyelination and MS

- Some articles with animals could be included in table 1. Emphasizing the effects on the immune system and not just on autophagy:

Li 2020 (10.3389/fneur.2020.590884)

Hou 2018 (10.1016/j.jneuroim.2018.08.012)

LaMothe 2018 (10.3389/fimmu.2018.00281)

Rezapour-Firouzi 2018, 2019, 2020a,b

2.1. Role of mTOR in innate immune responses in myeloid, microglial cells and 228 inflammasome

- Please include the recent publication: Borim 2022 (10.1089/jir.2021.0206).

4. Conclusion

- One sentence is confuse: "In the immune system, mTOR inhibition and increased autophagy can lead to increased antigen presentation and immune activation/T cell activation, and this is supported both by observations in animals and humans", lines 353-355. Is this in a condition of homeostasis? Because some studies show different scenario in inflammatory conditions, specially in EAE/MS. It would be interesting to clarify.

Author Response

Reviewer 1: The review is well written, succinct and brings important information in the context of mTOR in MS. Here are some small notes:

  1. Abstract: The abstract could be improved. I see no need to expose adverse events ("such as mouth ulceration and menstrual abnormalities") and talk more about the possible mechanism of action in the disease.

Response: We thank the Reviewer the kind words and are glad they found the review well-written, succinct, and carrying important information. We have edited the abstract according to the Reviewer’s suggestions and added more information regarding possible mechanisms by which of mTOR inhibitors can control disease activity. In addition, we have deleted the detailed description of temsirolimus adverse events. Changes can be found in lines 12-18 (page 1).

  1. Introduction: The authors focus only T cells, but in MS there is also the action of B cells and resident innate immune cells (microglia). The other cells (B and innate cells) are important in MS and are also affected by mTOR. Considering that the authors included these cells in the figure 2, in important to mentioned the function of these cells in MS pathogenesis.

Response: We agree and have added more information regarding the pathogenetic roles of B cells and microglia. Additions and changes can be found in the abstract, lines 12-18 (page 1) and in the introduction, lines 57-69 (page 2). Apart from the new information added, we wanted to point out that we had already touched upon the role of B and innate cells in the initially submitted manuscript (lines 51-57 (page 2), lines 250-265 (page 7), and lines 300-321 (page 8).

  1. For the chapter on mTOR inhibitors: Furthermore, the authors bring data from the experimental model (EAE), but do not include a description of models in the text. This is important to the reader, as there are different models (rat, C57BL/6, SJL, passive, active), with specific characteristics that mimic or not the conditions observed in patients.

Response: We agree and have added more information in regard the different EAE models. EAE details are now provided in Table 2 and additions and changes can be found in lines 166-169 (page 4) and lines 178-180 (page 5).

  1. For the chapter on “Specific role of mTOR on the biological process of autophagy and its 162 relevance to CNS demyelination and MS”: Some articles with animals could be included in table 1. Emphasizing the effects on the immune system and not just on autophagy:

Li 2020 (10.3389/fneur.2020.590884)

Hou 2018 (10.1016/j.jneuroim.2018.08.012)

LaMothe 2018 (10.3389/fimmu.2018.00281Rezapour-Firouzi 2018, 2019, 2020a,b

Response: All the above references were added to the revised Table 2.

  1. For the chapter on “Role of mTOR in innate immune responses in myeloid

d, microglial cells and inflammasome”: Please include the recent publication: Borim 2022 (10.1089/jir.2021.0206).

Response: We have added the reference to the revised Table 2.

  1. Conclusions: One sentence is confusing: "In the immune system, mTOR inhibition and increased autophagy can lead to increased antigen presentation and immune activation/T cell activation, and this is supported both by observations in animals and humans", lines 353-355. Is this in a condition of homeostasis? Because some studies show different scenario in inflammatory conditions, especially in EAE/MS. It would be interesting to clarify.

Response: We thank the reviewer for the constructive comment. We have now included a clearer discussion of the possible dual role of autophagy both as a protective homeostatic regulator in immune and neural tissue cells or as a factor of the pathologic mechanism of antigen presentation to CD4+ T cells resulting in autoimmunity. The added clarification can be found in lines 374-378 (page 9).

Reviewer 2 Report

The authors present a well-designed and clear review of molecular dynamics of mTOR pathways as their impact in the physiopathology of Multiple Sclerosis particularly. The paper reflects a comprehensive review of appropriate investigational and clinical attempts of translating these findings into therapy with oppositional results so far. The authors acknowledge they are exploring theoretical scenarios and that parts of this work borders in a speculative review. Nevertheless, the paper provides a good view of the topic which in fact is somewhat original. The figures are well illustrated and the tables informative. I believe the paper will be of interest to the readers of IJMS. 

A list of drugs or potential therapeutic molecules inhibiting mTOR pathways besides the ones addressed in your paper would add dimension to your otherwise rather comprehensive review. 

Author Response

Reviewer 2

The authors present a well-designed and clear review of molecular dynamics of mTOR pathways as their impact in the physiopathology of Multiple Sclerosis particularly. The paper reflects a comprehensive review of appropriate investigational and clinical attempts of translating these findings into therapy with oppositional results so far. The authors acknowledge they are exploring theoretical scenarios and that parts of this work borders in a speculative review. Nevertheless, the paper provides a good view of the topic which in fact is somewhat original. The figures are well illustrated and the tables informative. I believe the paper will be of interest to the readers of IJMS. 

  1. A list of drugs or potential therapeutic molecules inhibiting mTOR pathways besides the ones addressed in your paper would add dimension to your otherwise rather comprehensive review. 

Response: We thank the reviewer for the kind words and the constructive comments. We have now added a representative list of drugs as a table (Table 1)
